# A denaturation-free protocol for *in situ* visualization of short nuclear DNA sequences using padlock probes with rolling-circle amplification

Ryoyo Ikebuchi[ID][1]*, Lu Xi[1], Dimitra Bouri[1], Valeriia Svintytska[ID][1], Hanna Davies[2], Pawel Olszewski[2], Bozena Bruhn-Olszewska[2], Jan P. Dumanski[2], Ulf Landegren[1]*

1 Molecular Tools, Department of Immunology, Genetics & Pathology, Science for Life Laboratory, Uppsala University, Uppsala, Sweden, 2 Precision Medicine, Department of Immunology, Genetics & Pathology, Science for Life Laboratory, Uppsala University, Uppsala, Sweden

* ryoyo.ikebuchi@igp.uu.se (RI); ulf.landegren@igp.uu.se (UL)

## Abstract

We report an approach for *in situ* detection of genomic DNA sequences, where transiently opening DNA duplexes are captured by circularizing DNA strands – padlock probes – that lock in place in a sequence-specific manner through the action of a DNA ligase. Reacted probes, wound around their target strands, are then replicated by rolling-circle amplification for localized fluorescence detection. The technique serves to shorten assay time and enables detection of shorter specific DNA sequences compared to standard fluorescence in situ hybridization, FISH. Genomic sequences with thousands of locally repeated copies were detected in human leukocytes with greater than 99% efficiency and less than 0.15% false positives in just a few hours. Using a longer variant of the protocol targets of as little as 36 or 112 nt were visualized, albeit at lower efficiency and with a higher false positive rate. The technique of targeting sequences in duplex DNA using padlock probes is promising for both research and clinical diagnostics.

## Introduction

Analysis of variation in genomic DNA (gDNA), including translocations, deletions and insertions, copy number variants and single-nucleotide exchanges, are important both in research and in clinical routine. Recently developed genome editing technologies, such as using CRISPR-Cas9 [1], contribute to an increased demand for robust methods to detect gDNA alterations. Bulk or single-cell PCR and sequencing-based methods are sometimes appropriate for these purposes, however localized detection at cellular resolution is often required [2].

Fluorescence *in situ* hybridization (FISH) is widely used both in research and clinically to visualize sequence-specific chromatin domains *in situ*. Conventional

**Data availability statement:** All relevant data are within the manuscript and its Supporting Information files.

**Funding:** This work was supported by grants to U.L from the Swedish Research Council (https://www.vr.se/english.html) [2013-06023, 2014-02969, 2018-05895, 2018-06156, 2018-02943 and 2022-00570], the Swedish Foundation for Strategic Research [SB16-0046], VINNOVA [2019-01464], the Swedish Cancer Society [19 0384] and the Novo Nordisk Foundation (https://novonordiskfonden.dk/en/) [NNF22OC0080331]. It was also supported in part by grants from the Swedish Cancer Society (https://www.cancerfonden.se/om-oss/about) [2023-01940] and from the Swedish Research Council [20-0889-PjF] to J.P.D. The funders had no role in study design, data collection and analysis, decision to publish, or preparation of the manuscript.

**Competing interests:** The authors declare no competing interests.

DNA FISH is time-consuming and limited in sequence resolution, and there is a considerable cost for hybridization probes. High-definition DNA FISH (HD-FISH) and Oligopaint FISH have been developed to adress these limitations by using chemically synthesized oligonucleotide pools [3,4]. Despite these improvements, DNA FISH still requires harsh denaturation of double-strand DNA (dsDNA) by formamide, which is a known teratogen and not recommended for clinical use. FISH protocols are time-consuming and include one or two overnight probe hybridizations [5], where forma-mide competes with FISH probes as hydrogen donors, interfering with base pairing [6]. Since most commercial FISH probes have total target sequences of around 300–700 kb for specific and prominent visualization, insertions and deletions of less than 10 kb cannot be demonstrated [7,8].

Padlock probes may overcome both requirements for sequence specificity and for prominent detection signals [9]. A few words about these probes are in order. Pad-lock probes are circularizing oligonucleotide reagents that can be used for detecting either DNA or RNA, in solution or *in situ*, and reacted probes may be locally amplified through rolling circle amplification (RCA) [10–12]. Circularizing probes can also be used for localized analyses of proteins by allowing DNA-conjugated antibodies to give rise to circular DNA strands [13]. Target recognition via sequences located at both the 5' and 3' ends of padlock probes together allow detection of short target sequences of 30–50 nt, and the probes serve to distinguish single nucleotide variants by virtue of the requirement for correct hybridization by the juxtaposed end sequences of the padlock probes, taking advantage of the stringent substrate requirements of ligases [14–16]. The requirement for target recognition via a pair of target sequences endows padlock probes with target specificity similar to that of PCR, permitting detection of single copy targets in the complex human genome. Since only intramolecular probe ligation produces templates for RCA, the probes remain specific even in very high multiplex, whereas risks of crossreactive detection by noncognate primer pairs increases rapidly when more reactions are combined in PCR, precluding higher mul-tiplexing [17].

While padlock probes and RCA are in extensive use for *in situ* detection of both RNA and protein with padlock probes and RCA [11,13], robust, generally applica-ble protocols for padlock probe-based *in situ* detection of genomic DNA have been lacking. We previously developed a padlock probe-RCA assay for *in situ* detection of mitochondrial DNA, using a mechanism that involves preparation of single-stranded target sequences in genomic DNA through digestion with a restriction enzyme and then an exonuclease [14]. However, the efficiency of detection was unsatisfactory [18]. Other groups have reported greatly improved detection efficiency, reaching 60–90% for single-copy sequences of around 50 nt, either by complementing the reactions with pairs of peptide nucleic acid (PNA) reagents [19]or with Cas9 proteins for dsDNA target recognition by the padlock probes [20]. These assays require dedi-cated probe designs and involve additional steps to access dsDNA target sequences. This can increase cost and also restrict target sequence selection, because of the requirement for pyrimidine-rich target sequences for PNA binding, and for

protospacer adjacent motifs (PAM) for selective recognition by Cas9. Accordingly, there is a need for inexpensive *in situ* padlock probe assays that can efficiently target any gDNA sequences.

Here, we report a protocol for denaturation-free padlock probing (dfPP), which does not require formamide or other specific regents to access target sequences in dsDNA. The method was optimized for detecting large target sequences within a few hours of assay time or for revealing target regions of 150 nt or less with a longer assay time. The protocols can find applications in research as well as for clinical diagnostics.

## Results

### Padlock probe hybridization without standard denaturation of dsDNA

We initially investigated several conditions for *in situ* detection of gDNA sequences, by combining elements from previously published FISH [5] and padlock probe protocols. The studies led us to a rapid and convenient protocol that depends on DNA breathing in place of regular denaturation – dfPP. Cell samples are prepared through fixation with Carnoy's solution, dried, digested with pepsin, followed by simultaneous padlock probe hybridization and ligation at an elevated temperature, and then RCA and visualization (Fig 1A and Materials and Methods). Using this protocol with a padlock probe specific for DYZ1 (about 3,000 copies on chromosome Y (chrY) [21]), nearly all male peripheral blood mononuclear cells (PBMCs) exhibited distinct fluorescent dots, typically showing a one-dot-per-nucleus ratio (Fig 1B left and supporting S1 Fig).

Typical dsDNA denaturation protocols in FISH involve heating samples at around 70°C in a buffer containing formamide to disrupt hydrogen bonds between base pairs of gDNA [22]. We investigated denaturation by heating at 70°C in 70% formamide buffer for 3 min, before the padlock probe hybridization/ligation step. Under these conditions, almost all DYZ1 signals in the form of fluorescent dots were seen outside the nuclei (arrowheads in Fig 1B right and supporting S1 Fig). We also confirmed that prior denaturation at either low pH (5 M HCl) or high (0.2 M and 0.5 M NaOH) failed to generate any detectable products (Supporting S2 Fig). We investigated the effects of denaturation in greater detail using the male T lymphocyte cell line Jurkat. As there is no scientific consensus about the karyotype and within-culture mosaicism of Jurkat [23–25], we did not evaluate the numbers of fluorescence dots per nuclei. Nonetheless, we note that DYZ1 signals appeared as one or two dots in most Jurkat cells, using our dfPP protocol (Fig 1C). As previously seen, most signals generated using padlock probes after conventional denaturation of dsDNA were located outside the nuclei of the Jurkat cells (arrowheads in Fig 1C), while sparse signals inside nuclei were weaker than those generated by dfPP. No signals, either in- or outside-nuclei, were observed in the female cell line GM12878 (Fig 1D), indicating that the signals in Jurkat cells were indeed chrY-specific signals. The data demonstrate that standard dsDNA denaturation failed to result in optimal detection of gDNA regions by padlock probes with RCA amplification.

The *in situ* detection technique presented herein does not depend on outright denaturation of gDNA, as gently elevated temperatures suffices to allow target recognition by the padlock probes. Several other enzyme-assisted DNA assays require only modestly elevated temperatures for primers/probes to hybridize to their dsDNA targets. For example, the Wildfire isothermal amplification technique depends on "DNA breathing", allowing hybridization probes to access transitorily denatured gDNA regions at moderately elevated temperatures [26]. Also the loop-mediated isothermal amplification (LAMP) technique proceeds by continuously repeating primer binding to target DNA sequences transiently available through DNA breathing at approximately 60°C [27,28].

We investigated the effect of increasing hybridization/ligation temperatures as a means to access target sequences in gDNA. Although some of the tested temperatures were above the calculated annealing temperatures of the DYZ1-specific padlock probe (52.3 and 52.9°C at the 5'- and 3'-ends, respectively), all investigated temperatures between 45 and 65°C generated chrY-specific signals in Jurkat cells (Fig 1E; Supporting S3 Fig for 50 and 60°C). However, hybridization and ligation at 65°C yielded more than two signals in individual nuclei more often than reactions at 45 or 55°C, indicating poor

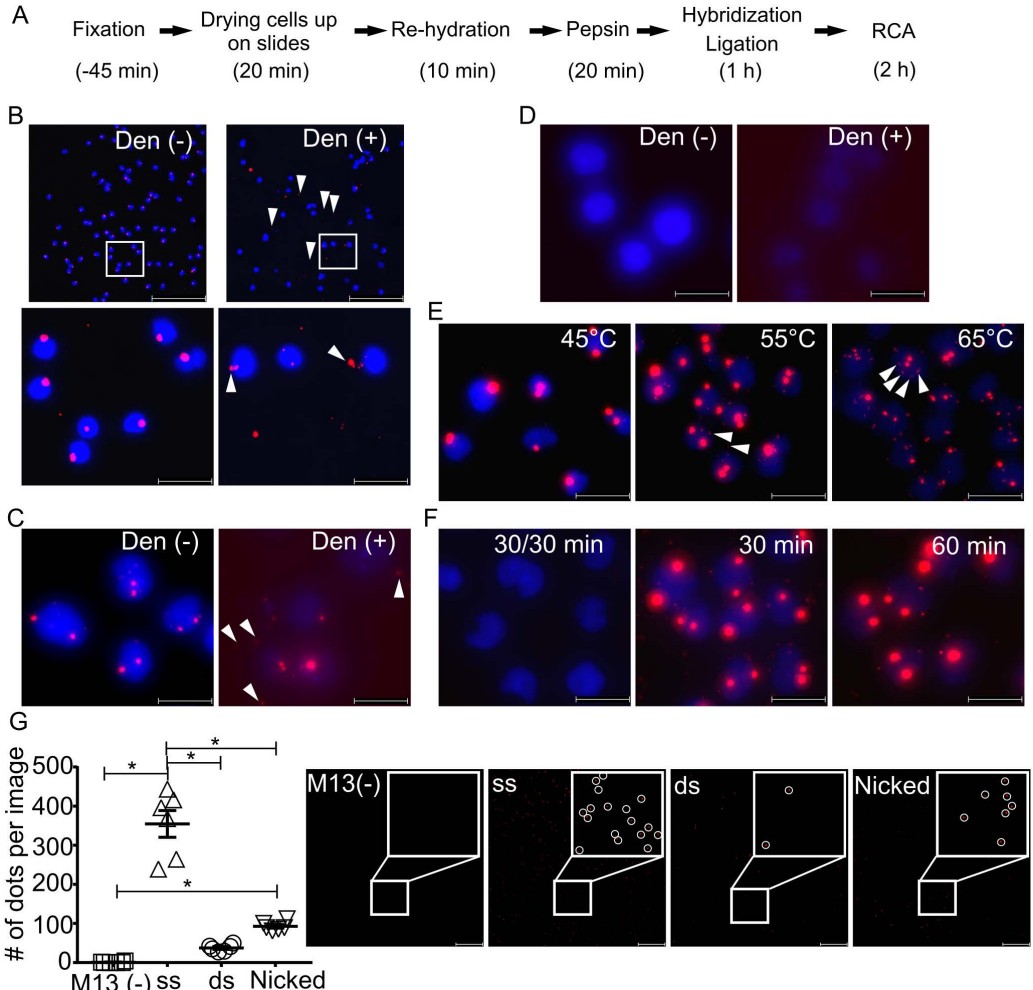

**Fig 1. Investigation of the mechanism of dfPP.** (A) Workflow of dfPP. (B, C and D) Detection of DYZ1 (3,000 copies in chrY, red signals) in male PBMC (B), male Jurkat cells (C) and female GM12878 cells (D) without or with prior standard denaturation (Den – or Den +, respectively). The areas demarcated by white squares in the top images are shown at greater magnification below. White arrow heads indicate RCA signals located outside nuclei. (E) Detection of DYZ1 in Jurkat cells with combined padlock probe hybridization and ligation at the indicated temperatures, followed by RCA. White arrow heads indicate smaller RCA signals than the main signals. (F) Detection of DYZ1 in Jurkat cells by padlock probe hybridization for 30 min, followed by washing and then ligation for 30 min (left, 30/30 min) or by simultaneous hybridization and ligation for either 30 or 60 min (middle and right, 30 min and 60 min). (G) Detection of ssDNA (ss) and dsDNA (ds) forms of the M13 phage genome immobilized on slides. A nick was induced in the supercoiled circular dsDNA M13 by the nickase Nb.BtsI (Nicked). Samples omitting any types of M13 were treated as negative controls (M13 -). The white circles in enlarged images indicate fluorescent dots. The dot plot indicates numbers of fluorescent dots (RCA products) per a microscopy field (n = 6). Asterisk indicates $p < 0.05$. Scale bars in (B) and (G) and in other magnified images represent 100 µm and 20 µm respectively. Nuclear DNA is stained in blue except for in (G).

localization of detection reactions, presumable due to displacement of either target sequences, circularized probes and/or RCA products at the higher temperature of 65°C.

To investigate the dfPP mechanism we compared *in situ* detection of the DYZ1 motif via simultaneous hybridization and ligation using the dfPP protocol versus a variant of this procedure where a first hybridization step was followed by a subsequent ligation reaction. Combined hybridization and ligation for either 30 min or 1 h resulted in efficient DYZ1 detection in male Jurkat cells, while separate 30 min hybridization followed by a rinse before a 30 min ligation failed to generate any signals (Fig 1F).

In order to further explore the dfPP detection mechanism we compared the numbers of RCA products derived from padlock probes incubated at 55°C with a DNA ligase and three forms of DNA from the M13 bacteriophage serving as ligation templates; ssDNA, the replicate form dsDNA, and dsDNA that we had first treated with a nicking endonuclease to avoid supercoiling. RCA was primed by a biotin-modified oligonucleotide to anchor RCA products on streptavidin-coated slides for imaging by microscopy. As expected, the ssDNA ligation template yielded the greatest number of RCA products (350 ± 34/microscopy image) (Fig 1G). However, circular M13 dsDNA still yielded 37 ± 3.6 RCA products, while the nicked target resulted in 92.7 ± 4.65 products, indicating that it is possible for padlock probes combined with ligases to recognize dsDNA templates without prior denaturation.

These data suggest a model whereby transient hybridization by the 5' and 3' ends of the padlock probes to breathing target sequences is rendered permanent by enzymatic circularization of padlock probes, irreversibly trapping the probes in place, wound around their target gDNA strands. Despite being wound around one strand in a DNA duplex the reacted padlock probes can still serve as templates for RCA [19,29] (Fig 2). Feng et al. have suggested that the DNA unstacking or breathing effect may be enhanced by hydrophobic residues in DNA enzymes [30], perhaps contributing to the DNA duplex invasion.

gDNA (red) exhibits thermally driven structural fluctuations at elevated temperatures, also referred to as DNA breathing or transient opening of the DNA duplex. The action of a DNA ligase serves to irreversibly capture temporarily a hybridizing padlock probe (blue) as a DNA circle, topologically linked to the DNA duplex. The padlock probe, still wound around its target strand, can next serve as a template for RCA, generating a single stranded product (green) [19,29].

### *In situ* detection of a single copy gDNA target

Localized detection of short sequences in gDNA can have great value to trace the distribution of, e.g., integrated viral genomes in tissues and to characterize translocations, insertions and deletions. Given sufficient sensitivity and selectivity dfPP could also provide a window into somatic sequence variation and clonal evolution of tissues. dfPP combines the proven ability of padlock probes to resolve single-copy sequences in complex genomes while distinguishing single-nucleotide variation, with prominent localized DNA amplification signals through RCA. Fluorophore-labeled probes can hybridize to the hundreds of complements of the probes that form by RCA for enhanced detection [13,14,31,32].

We investigated the efficiency of dfPP for detecting single-copy sequences by extending the incubation time for RCA from 2 h (Fig 3A) to overnight (Fig 3B) following another RCA-based gDNA detection method [19]. We also introduced a dry-aging step at 65°C for 1 h before re-hydration and probing of the samples (Fig 3C). This treatment is a commonly used to improve detection efficiency in cytogenetic FISH protocols [33]. We used this slightly modified dfPP protocol to target a 36 nt single-copy sequence in the chrY-specific *TMSB4Y* gene. Detection efficiencies varied between 13.9 and 62.0% (38.08% ± 19.20% on average) in cells from five male donors (Fig 3D and supporting S4 Fig). In cells from four female

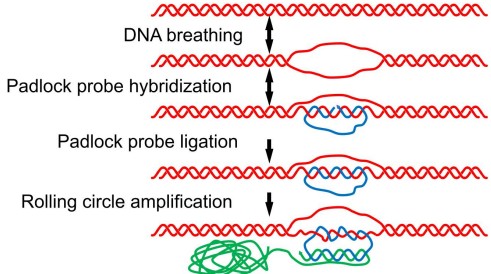

**Fig 2. A model for the mechanism of dfPP.**

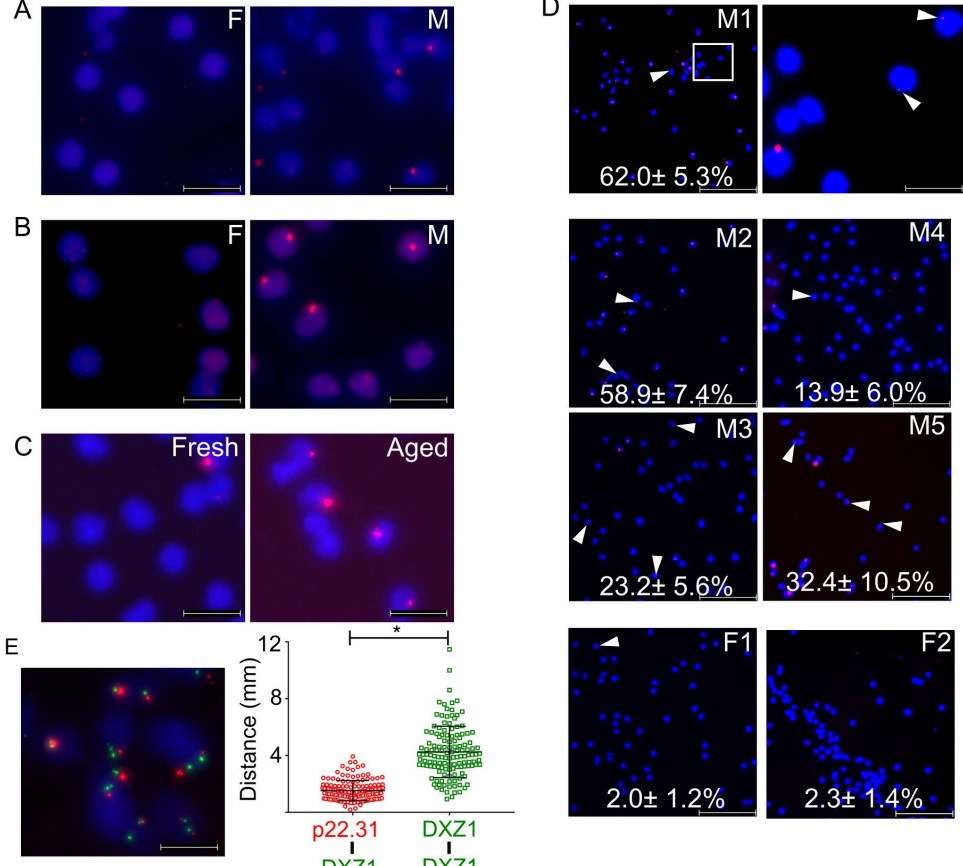

**Fig 3. Detection of a single copy gDNA target using dfPP. (A and B)** A 36 nt single copy sequence in the *TMSB4Y* gene (red) located in chrY was probed for in female (F) or male **(M)** PBMCs with RCA for 2 h (A) or overnight **(B)**. **(C)** Detection of the *TMSB4Y* target sequence (red) in fresh male PBMC (fresh) or after dry-aging at 65°C (aged). **(D)** Detection of *TMSB4Y* (red) at the indicated efficiencies in PBMC from 5 male (M1-5) and 2 female donors (F1-2) by a protocol with dry-aging and overnight RCA. The area bordered by a white square in the left image for male donor 1 is shown at greater magnification in the right image. The proportions in some images of female or male cells indicate the observed false positive rate and detection efficiencies, respectively. **(E)** Detection in female SKBR3 cells of a p22.31 sequence present in 11 copies (red) and DXZ1 (green), both located in chrX. The dot plot indicates the measured distances between the closest pairs of signals for p22.31 and DXZ1 (left) and for pairs of DXZ1 signals (right) in individual nuclei. The asterisk indicates that $p < 0.05$. Scale bars in (D) and in other magnified images represent 100 µm and 20 µm respectively. Nuclear DNA is stained in blue.

donors a false positive rate of 2.85% ± 1.08% (n = 4) was seen. Individual RCA signals varied in intensity, indicating that the extent of RCA was variable (white arrow heads in Fig 3D).

To evaluate the detection efficiency for multi-copy targets, we analyzed a repeated sequence in the chrX p22.31 region (11 copies), which is undetectable by conventional FISH assays. The detection efficiency at p22.31 was 51.6% ± 6.8% (Fig 3E). Furthermore, we confirmed the specificity of this variant of dfPP by observing proximity between signals when targeting the centromeric repeated region DXZ1 in chrX (about 1,000–1,500 copies [34]) and a repeated sequence in the chrX p22.31 region (11 copies), located at a distance of about 50 Mbp in chrX.

### *In situ* detection of tandem repeated DNA sequences

To demonstrate additional advantages of dfPP, namely its short assay time and precise target detection, we analyzed high-copy number targets, including the heterochromatic repeated region DYZ1 and a palindromic sequence Pal3 (about

178 copies on chrY) [35], using the same dfPP protocol in Fig 1. Both regions were visualized only in male PBMCs but absent in female cells (Fig 4A and B). In some male cells, these chrY-specific signals were observed outside nuclei when using brighter imaging settings (Fig 4A, lower images), indicating displacement of either target genes, ligated padlock probes or RCA products. Detection efficiencies for DYZ1 and Pal3 were 99.6%±0.3% and 96.2%±1.6% of the cells, respectively, and false positive proportions were 0.14%±0.20% and. 0.28%±0.39%. We note that these values are similar to or better than reported values for a commercial FISH assay targeting DYZ1 for *in vitro* diagnostic use (07J20-050, Abbott), which are 98.90% and 0.014% for 71 male and 57 female bone marrow specimens, respectively.

We also simultaneously detected DYZ1 in chrY and DXZ1 in chrX. These sequences represent established diagnostic targets for FISH assays for evaluating sex chromosome disorders. As expected, female PBMCs showed two DXZ1 dots in individual nuclei, while male cells exhibited one DXZ1 and one DYZ1 signal (Fig 4C). RNase treatment did not alter either

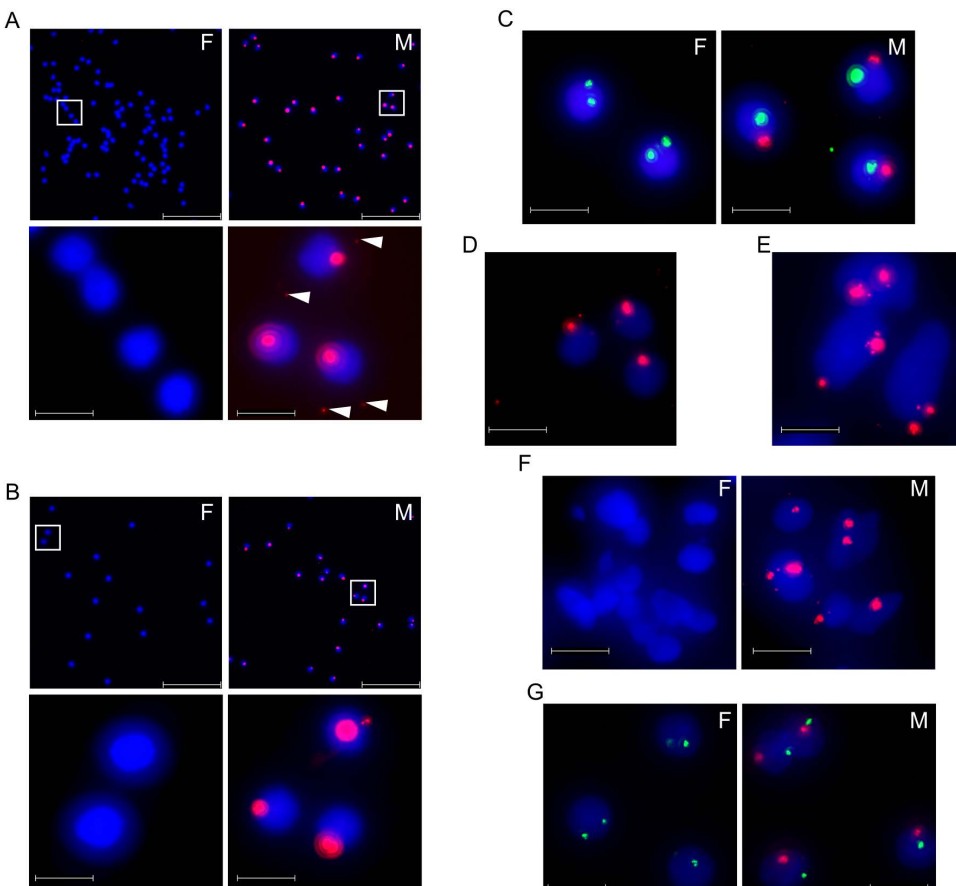

**Fig 4. gDNA detection for high copy-number targets with dfPP.** (A-G) Representative images of RCA signals from triplicate assays for the target regions, DXZ1 (present in 1,000–1,500 tandem copies in chrX), DYZ1 (3,000 tandem copies in chrY) and the palindromic sequence region Pal3 (178 copies in chrY) in male (M) and female (F) cells (n = 5 and 4 for PBMCs and n = 3 for cell lines). (A) Detection of DYZ1 in PBMCs (red signals). The areas demarcated by white squares in the top images are shown at greater magnification below. White arrow heads in the lower images indicate weak RCA signals located outside nuclei. (B) Detection of Pal3 (red) in PBMCs. Images at greater magnification are shown below. (C) Detection of DYZ1 (red) and DXZ1 (green) in PBMCs. (D) Detection of DYZ1 (red) in male PBMCs after RNase treatment. (E) Detection of DXZ1 (red) in female SKBR3 cells. (F) Detection of DYZ1 (red) in female and male brain sections (n = 2 respectively). (G) Detection of DYZ1 (red) and DXZ1 (green) in PBMCs with a shorter, 2½ hr dfPP protocol. Scale bars in (A) and (D) and in other magnified images represent 100 µm and 13.3 µm respectively. Nuclear DNA is stained blue in all panels.

signal intensity or detection efficiency (Fig 4D), confirming that the signals represented true DNA target sequences. This variant of the dfPP procedure was also applied for adherent cells and for tissue sections by visualizing DXZ1 and DYZ1 in breast cancer cell lines (Fig 4E) and in brain tissue, respectively (Fig 4F). The above probes targeted 36 nt sequences present in hundreds to thousands of copies in chrX or chrY. These data indicated that dfPP can visualize gDNA regions of at least 6–7 kb (178 copies of 36 nt target sequences). This target size is shorter than that of typical target regions for conventional FISH probes [7,8], while still offering greater than 95% detection efficiency.

Next, we attempted to reduce the assay time of dfPP for rapid detection of genomic sequences in order to simplify workflow and allow rapid response to requests for analysis [36]. DYZ1 and DXZ1 were specifically detected using a rapid version of the dfPP protocol with padlock probing and RCA for 20 min each (Fig 4G). The total assay time from fixation to mounting is two and a half hr, much shorter than FISH assays that typically require 1–3 days of probe hybridizations [22].

While there is room for improvement to increase detection efficiency and reduce false positive signals for single copy detection, the results illustrate the potential of dfPP to detect short region in gDNA *in situ* while omitting the conventional denaturation of dsDNA.

## Discussion

Standard FISH protocols generally require target gDNA sequences larger than 10–100 kb for unambiguous detection with published protocols [7,8] and 300–700 kb using commercial FISH reagents. There is, however, a growing demand for higher resolution to detect limited insertions and deletions in inherited or malignant disease, as well as to evaluate results of genome editing in targeted cells. Here we developed a padlock probe-based protocol for *in situ* gDNA detection, taking advantage of the mechanism of DNA breathing (Table 1). Compared to regular FISH, our protocol allows detection of much shorter target sequences, and the technique can be applied for any target sequences with no need for specific target sequence features. Other techniques such as PNA-RCA-FISH [19], CASFISH [37] and CasPLA [20] can also detect short target sequences but target selection is constrained by sequence requirements imposed by PNA probes and by Cas9, respectively. By avoiding a denaturation step, the mild dfPP procedure presented herein may prove advantages for simultaneous analyses of other, sensitive molecules in the samples.

We report herein a protocol with variation for *in situ* detection of specific gDNA sequences. Repeated regions of modest size can be visualized in a rapid and convenient procedure with satisfactory efficiency. We demonstrated detection of a single copy target with an efficiency that ranged between 15 and 60% per cell. Single copy detection was associated with some risk for false positives, as observed in other gDNA detection technologies (Table 1), that could limit applications for genotyping single copy target sequences depending on the context. It is perhaps not surprising that the detection efficiency by a single padlock probe is less than 100%. The efficiency of padlock probe-based detection of double-stranded

**Table 1. Comparison of *in situ* gDNA detection assays.**

| | Minimum target length (efficiency) | Target length with >90% efficiency | Target sequence requirements | Assay duration |
|---|---|---|---|---|
| dfPP | 36 nt (13–65%, 38% on average)* | 6,408 bp [36 bp × 178 copies] | Any | <2.5 h (2 days for short target sequences) |
| FISH | 10,000–100,000 bp [7,8] | 300,000–700,000 bp | Any | 2 days |
| CASFISH | <1,035 bp [23 bp× <45 copies] (60-80%) | Not tested | Near protospacer adjacent motif for Cas9 binding | 20 min – 1.5 h |
| PNA-RCA-FISH | 22 bp (90%)* | | Between a pair of pyrimidine sequences for PNA binding | 2 days |
| CasPLA | 40 bp [two single guide RNA recognition sites] (60%)* | Not tested | Between a pair of Cas9-recognition sequences | 1 day |

* With a few percent false positives

M13 DNA was lower than for the single stranded form of the bacteriophage DNA. The <62% detection efficiency was reported for single stranded mRNA detection by expansion sequencing with maximum 16 padlock probes [38], indicating less than 5% efficiency per padlock probe. For *in situ* detection of gDNA it is also conceivable that specific target sequences may be unavailable in fixed samples, or that the local chromatin may be too constrained to undergo the DNA breathing required for dfPP.

Once successfully circularized, the padlock probes next must serve as templates for RCA in order to be detected. We found that the sizes of individual RCA products resulting from padlock probe-based gDNA detection were variable, and they tended to be smaller and less uniform than in other in situ RCA applications. In order to detect products of single probe molecules it was necessary to perform RCA overnight (−16h). Nonetheless, some signals remained weak as illustrated in the figures. By contrast, one hour replication yields easily detected individual RCA products arising from ssDNA circles immobilized on slides [39] or from circularized padlock probe having recognized mRNA sequences *in situ* [11] or from protein detection via in situ proximity ligation assays [13]. We note that Yaroslavsky and Smolina also resorted to overnight RCA for *in situ* gDNA detection using padlock probes of a similar size [19] and Zhang et al. observed smaller RCA products when targeting nuclear dsDNA rather than mitochondrial dsDNA [20]. These reports and our results herein suggest that RCA may be compromized in nuclei.

Detection efficiencies lower than 100% may still serve the purpose of detecting local clones of mutant cells [40]. Moreover a reduced detection efficiency for single copy DNA sequences by dfPP may be alleviated by targeting more tumor-specific somatic mutations in parallel, taking advantage of the suitability of padlock probing for multiplexing. Nonetheless, further technology development is warranted to improve assay performance. Scalable in situ gDNA analyses through dfPP can find applications in basic research in genome structure as well as for clinical applications in medical genetics and for characterizing genetic heterogeneity in tumor tissues.

In conclusion, we demonstrated padlock probe assays with RCA readout for *in situ* gDNA detection. The protocols can extend the scope, and reduce the time required for *in situ* detection of gDNA sequences in basic research and for clinical diagnostics.

## Materials and methods

### Cells and tissues

Jurkat (TIB-152, ATCC) and GM12878 cells (Coriell Institute for Medical Research) were cultured at 37°C in RPMI1640 (11875093, Gibco), supplemented with 1% penicillin/streptomycin/glutamine (12090216, Gibco) and 10% fetal bovine serum (11560636, Gibco). For SKBR3 cells (HTB-30, ATCC), the above medium was supplemented with 20mM HEPES (15630080, Gibco; pH 7.2–7.5) and 1mM sodium pyruvate (11360070, Gibco). Human peripheral blood was collected in BD Vacutainer® CPT™ Mononuclear Cell Preparation Tubes (362781, BD), and PBMCs were isolated according to the manufacturer's instructions. The tubes were centrifuged within 4h of collection; the PBMC fraction was washed with PBS and cell numbers and viability were determined using Countess II FL automated cell counter (Thermo Fisher). To avoid using samples where some cells might have lost chrY with aging [41], we selected male donors who were under 50 years old. Human blood and brain tissues were recruited in 2018–2019, obtained from Uppsala University Hospital, Sweden, and handled in accordance with ethical permission by Regional Ethics Review Board in Uppsala (2015/092). Written informed consent was obtained from all individual participants included in the study.

### DNA oligonucleotides and padlock probe design

All DNA oligonucleotides were synthesized by Integrated DNA Technologies and their sequences are shown in Supporting S1 Table. Padlock probes were chemically modified with 5'-phosphates and purified by polyacrylamide gel electrophoresis. To select target sites for the probes, we used CRISPRdirect software [42] in order to identify unique sequences using sequence information registered at MSY Breakpoint Mapper [35] for chrY and the UCSC Genome Browser [43] for other

chromosomes. The probes were designed with 18–23 target-complementary nucleotides at both the 5'- and 3'-ends. To increase ligation specificity for targets present in fewer than 20 copies, the target-complementary sequences at the 5' and 3' ends were designed to differ in melting temperatures by less than 10°C [44]. We checked for regions similar to the target sequences in a total human genome database (hg38) by Human BLAT search [45] (but not BLAST [46]). Hybridization sites for RCA primers and detection oligonucleotides were analyzed by NUPACK [47] to avoid secondary structures.

## Cell preparation on slide glasses

Suspension cells were fixed using 10 ml freshly-prepared pre-chilled Carnoy's solution (methanol (34860, Sigma-Aldrich) and glacial acetic acid (338826, Sigma-Aldrich) at a ratio of 3:1) at −20°C in 15 ml tubes for 10 min, followed by centrifugation and then two additional fixations in 5 ml Carnoy's solution. The cells were suspended in Carnoy's solution ($5 \times 10^5$ cells/40 µl) and dropped onto ImmunoSelect Adhesion Slides (SQ-IS-10100, Dianova) from about 3−5 cm, followed by drying for 20 min in a humidity box (50−80% relative humidity at 22−25°C). Adherent cells and tissue sections were fixed three times with 20 ml Carnoy's solution in slide glass holders and dried in 40 µl Carnoy's solution under the condition described above. The 5 µm tissue sections were fixed immediately after sectioning and attaching them to ImmunoSelect Adhesion Slides while avoiding drying before fixation. The fixed cells were subjected to a rehydration step or were stored at −20°C in Carnoy's solution for up to two days for target sequences <150 bp, and up to a few months for larger target regions. This preparation step is critical for performing the dfPP assay. Great attention was given to maintaining relative humidity also after opening the lid of the humidity box for dropping cell suspensions.

## Hybridization, ligation and RCA

Unless otherwise indicated, the standard protocol for detecting target sequences with >100 tandem repeated copies was as follows: the fixed cells were rehydrated in 2×SSC buffer in Grace Bio-Labs SecureSeal hybridization chambers (GBL621502, Merck) for 10 min at 37°C, followed by protein digestion using pepsin (P7000, Sigma-Aldrich) at 30 (PBMCs and brain tissues), 100 (Jurkat and GM12878 cells), or 900 (SKBR3 cells) µg/ml in 100 mM HCl (1.00314.1000, Merck) for 20 min at 37°C. The slides were washed in 2×SSC buffer twice and rinsed with buffer A (0.1 M Tris-HCl pH 7.5, 0.15 M NaCl and 0.05% Tween20 (P9416, Sigma-Aldrich)). Padlock probes were hybridized to target sequences in the cells and in the same reaction enzymatically circularized in a ligation mixture of 200 µg/ml bovine serum albumin (B9000, New England Biolabs), 100 µg/ml salmon sperm DNA (15632−011, Thermo Fisher), 30 mM NaCl, 100 nM padlock probes and 1.6 U/µl Taq ligase (M0208, New England Biolabs) in 1×Taq ligase buffer for 1 h at the temperature described in S1 Table. Holes of the SecureSeal were covered by sealing tape (95.1994, Sarstedt) to avoid evaporation of the ligation solution. After washing once with buffer B (2×SSC buffer with 0.05% Tween20) at 37°C, circularized probes were amplified in an RCA mixture (200 µg/ml bovine serum albumin, 1 mM DTT (A39255, Thermo Fisher), 250 µM dNTP (R0181, Thermo Fisher), 1 µM RCA primer 1, 25 nM fluorophore-conjugated detection oligonucleotide (Texas-red or Alexa Fluor-594-conjugated detection 1 and/or Alexa Fluor-488-conjugated detection 2), 25 nM Hoechst 33342 (62249, Thermo Fisher) and 0.5 U/µl phi29 DNA polymerase (4002, Monserate) in 1×phi29 buffer) for 2 h at 37°C. The cells were washed twice with buffer B and mounted with SlowFade Gold Antifade (S36936, Thermo Fisher). For tissue sections, autofluorescence was reduced using TrueView autofluorescence quenching (Vector Laboratories) according to the manufacturer's protocol before mounting. To prove assay specificity for DNA (Fig 1e), RNA was in some experiments digested with RNase A (1007885, Qiagen) at 200 µg/ml for 30 min at 37°C during rehydration of cells in 2×SSC buffer.

For detecting target sequences <150 bp, samples on microscope slides were kept at 65°C for 1 h as a dry-aging step before rehydration with 2×SSC buffer. Padlock probe hybridization and ligation was performed for 2 h, followed by RCA for 16–20 h at 30°C with the RCA mixture omitting detection oligonucleotide and Hoechst 33342. RCA products were subsequently stained for 15 min at 37°C with 25 nM detection oligonucleotides in 2×SSC, supplemented with 200 µg/

ml bovine serum albumin, 100 µg/ml salmon sperm DNA, and 25 nM Hoechst33342. After two washes with buffer B, the samples were mounted using SlowFade Gold Antifade.

## Imaging

Stained cells were imaged by epifluorescence microscopy with a DMi8 microscope (Leica) equipped with a DFC9000 sCMOS camera (Leica), excitation and emission filters for DAPI, FITC and Texas Red, and with objective lenses HC PL APO 20x/0.75 and HC PL APO 40x/1.30. We collected three images per sample using the LAS X software (Leica), while imaging 25–100 nuclei per image and avoiding obvious debris. Maximally projected z-stacks of 7–30 slices were exported as tiff files. The numbers of nuclei in the tiff images were measured using CellProfiler software [48,49]. In most experiments, RCA products were counted manually. The numbers of RCA products was then divided by the number of nuclei to calculate rate of false and correct positive signals in negative and positive samples, respectively. Distances between RCA products were estimated manually with ImageJ [50].

## Hybridization, ligation and RCA for detecting M13 DNA

In order to relax the supercoiled dsDNA, M13mp18 RF I DNA (N4018S, New England Biolabs) was nicked with Nb. BtsI (R0707S, New England Biolabs) according to the manufacturer's protocol and stored at −20°C until use. One nM ssDNA M13 (N4040S, New England Biolabs), dsDNA supercoiled or nicked M13 DNA, serving as ligation targets, were incubated with 10 nM padlock probes, 0.8 U/µl Taq ligase and a biotin-conjugated RCA primer 2 (30 nM) at 55°C for 2 h. The reactions were diluted 200-fold with buffer C (5 mM Tris-HCl pH 7.5, 0.5 mM EDTA (15575020, Thermo Fisher), 1 M NaCl and 0.05% Tween20) and added to streptavidin-coated slides (custom-made by TRIDIA) with FlexWell incubation chambers (204916, Grace Bio-Labs) for 1 h at room temperature. The slides were rinsed twice with buffer C, followed by addition of 250 µM dNTP, 0.2 U/µl phi29 DNA polymerase and 15% PEG4000 (supplemented with T4 DNA ligase (EL0012, Thermo Fisher)) in 1 × phi29 buffer for RCA at 37°C for 1.5 h. After two washes, RCA products were stained for 10 min at 37°C with 25 nM Alexa Fluor-594-conjugated detection 3 in buffer C, supplemented with 10% dextran sulfate (S4030, Millipore). After two washes with buffer C for 5 min, the samples were mounted using SlowFade Gold Antifade. RCA products were visualized with the objective lens HC PL APO 20x/0.75 and counted with CellProfiler software as described previously [39].

## Statistical analyses

Mann-Whitney tests and one-way ANOVA with Tukey's tests were applied using GraphPad Prism version 7.05 (GraphPad Software). Data is presented as mean ± SEM of replicate determinations.

## Supporting information

**S1 Fig. Detection of DYZ1 without or with standard denaturation.** Detection of DYZ1 in male PBMCs without or with prior standard denaturation (Den – or Den + , respectively). The composite images consists of four panels: nuclear DNA staining (top left), RCA signal (top right), bright-field microscopy (bottom left), and the merged image (bottom right).White arrow heads indicate RCA signals locating clearly outside nuclei.
(TIF)

**S2 Fig. Detection of DYZ1 without or with denaturation at low/high pH.** Detection of DYZ1 in male PBMCs without (Den –) or with prior denaturation by 5 M HCl, 0.2 M NaOH or 0.5 M NaOH, respectively. Scale bars represent 20 µm respectively.
(TIF)

**S3 Fig. Detection of DYZ1 with dfPP at various temperatures.** Detection of DYZ1 in Jurkat cells with combined padlock probe hybridization and ligation at the indicated temperatures, followed by RCA. White arrow heads indicate smaller RCA signals than the main signals. Scale bars represent 20 μm respectively.
(TIF)

**S4 Fig. Detection of TMSB4Y with dfPP.** Detection of TMSB4Y in male PBMCs (M1). The composite images consists of four panels: nuclear DNA staining (top left), RCA signal (top right), bright-field microscopy (bottom left), and the merged image (bottom right). White arrow heads indicate smaller RCA signals than the main signals.
(TIF)

**S1 Table. S1 Table PLOSOne.**
(XLSX)

## Author contributions

**Conceptualization:** Ryoyo Ikebuchi, Ulf Landegren.

**Data curation:** Ryoyo Ikebuchi, Lu Xi.

**Formal analysis:** Ryoyo Ikebuchi.

**Funding acquisition:** Jan P. Dumanski, Ulf Landegren.

**Investigation:** Ryoyo Ikebuchi, Dimitra Bouri, Valeriia Svintytska, Hanna Davies, Pawel Olszewski, Bozena Bruhn-Olszewska.

**Methodology:** Ryoyo Ikebuchi, Dimitra Bouri, Valeriia Svintytska.

**Project administration:** Ryoyo Ikebuchi, Jan P. Dumanski, Ulf Landegren.

**Resources:** Ryoyo Ikebuchi, Hanna Davies, Pawel Olszewski, Bozena Bruhn-Olszewska, Jan P. Dumanski.

**Supervision:** Ryoyo Ikebuchi, Ulf Landegren.

**Validation:** Ryoyo Ikebuchi, Dimitra Bouri, Valeriia Svintytska.

**Visualization:** Ryoyo Ikebuchi.

**Writing – original draft:** Ryoyo Ikebuchi, Ulf Landegren.

**Writing – review & editing:** Ryoyo Ikebuchi, Lu Xi, Bozena Bruhn-Olszewska, Jan P. Dumanski, Ulf Landegren.

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
