## [Decision Letter · Decision Letter 0]

4 Sep 2025

Dear Dr. Ikebuchi,

Thank you for submitting your manuscript to PLOS ONE. After careful consideration, we feel that it has merit but does not fully meet PLOS ONE’s publication criteria as it currently stands. Therefore, we invite you to submit a revised version of the manuscript that addresses the points raised during the review process.

We look forward to receiving your revised manuscript.

Kind regards,

Ruijie Deng

Academic Editor

PLOS ONE

Journal Requirements:

 “This work was supported by grants to U.L from the Swedish Research Council (https://www.vr.se/english.html) [2013-06023, 2014-02969, 2018-05895, 2018-06156, 2018-02943 and 2022-00570], the Swedish Foundation for Strategic Research [SB16-0046], VINNOVA [2019-01464], the Swedish Cancer Society [19 0384] and the Novo Nordisk Foundation (https://novonordiskfonden.dk/en/)[NNF22OC0080331]. It was also supported in part by grants from the Swedish Cancer Society (https://www.cancerfonden.se/om-oss/about) [2023-01940] and from the Swedish Research Council [20-0889-PjF] to J.P.D.”

Reviewers' comments:

Reviewer's Responses to Questions

**Comments to the Author**

1. Is the manuscript technically sound, and do the data support the conclusions?

Reviewer #1: Yes

Reviewer #2: No

Reviewer #3: Partly

2. Has the statistical analysis been performed appropriately and rigorously?

Reviewer #1: Yes

Reviewer #2: No

Reviewer #3: No

3. Have the authors made all data underlying the findings in their manuscript fully available?

Reviewer #1: Yes

Reviewer #2: No

Reviewer #3: No

4. Is the manuscript presented in an intelligible fashion and written in standard English?

Reviewer #1: Yes

Reviewer #2: No

Reviewer #3: No

Reviewer #1: This paper presents a novel genomic DNA FISH method based on padlock probe (PP) ligation and rolling circle amplification (RCA). PP coupled with RCA is a classic nucleic acid amplification method that has been applied to many fields. Recently, PP and RCA have been widely used in imaging-based spatial transcriptomics and have shown great advantages. However, for applications in genomic DNA in situ detection, PP coupled with RCA has always been hindered by low detection efficiency. Although the current study has completely solved this problem, it did provide some useful information to push forward applications of PP and RCA for in situ DNA detection. Below are a few questions:

1. In Lines 113-115, the authors suggested the RCA products outside the nuclei may be due to displacement. Although this may be true, is it possible that these RCPs are non-specific ligation products? Is there any way to test this?

2. In Line 178, is the HCl 5 M or 0.5 M? 5 M seems to be a very high concentration.

3. Have you tested a longer assay time to achieve higher detection efficiency for low copy number loci?

Reviewer #2: This manuscript by Ikebuchi and coworkers describes a padlock probe-based approach for in situ detection of genomic DNA sequences. Unfortunately, I am not enthusiastic about the significance of the findings described in this manuscript. The authors claim that their method is able to (1) shorten assay time, (2) detect much shorter target sequences, (3) detect any target sequence. However, the experimental data provided are too limited to robustly support these claims. I do not think this work is suitable for PLOS One.

Major comments:

1. The fluorescence images are of low quality. The overall experimental data of this work is very thin, and most of them are fluorescence images. Most importantly, all of the fluorescence images are blurry.

2. The main text figures do not include bright-field (BF) images to confirm the morphology of the cells during imaging. It would strengthen the presentation to incorporate BF images and their corresponding fluorescence/BF merged images directly into the main figures to clearly demonstrate that the method is tested and functional in cells. This addition would enhance the readers' confidence to ensure that the cells are not getting damaged during the labeling.

3. In Figure 2f, it is difficult to distinguish the fluorescent signals from the fluorescence images, so it is hard to evaluate whether the authors' statistics are accurate.

4. The placement of the schematic diagram in Figure 3 diminishes its effectiveness. For clarity, it would be more appropriate to present this figure earlier in the manuscript (e.g., as Figure 1). This would provide readers with a necessary overview of the experimental design before they encounter the detailed results, thereby improving the logical flow and comprehension.

5. The authors can use a simpler way of communicating their work here and without coming up with random words/terminologies. For example, “sequence resolution” in line 32.

Minor comments:

6. The manuscript itself is difficult to understand due to numerous grammatical errors.

7. Some fluorescence images lack scale bars.

Reviewer #3: In this manuscript, the authors describe a denaturation-free method for in situ genomic DNA detection based on padlock probes and rolling-circle amplification (RCA). Short-lived openings in the DNA duplex are captured by circularizing oligonucleotides—padlock probes—that are enzymatically sealed by a DNA ligase in a sequence-specific manner. Once circularized and coiled around their target strands, the probes are amplified via RCA, yielding localized fluorescent signals for high-resolution visualization. Using this approach, genomic regions with thousands of repeats were detected in human leukocytes within a few hours, achieving over 99% efficiency and below 0.15% false positives. I think this work would meet the criteria of PLOS ONE. if the following issues are properly addressed.

1.Regarding the organization of the manuscript, Figure 1 presents actual sample detection results, while Figure 2 shows methodological validation. It is recommended to place the content of Figure 1 after Figure 4.

2.In Figure 2, the validation of DYZ1 specificity is not sufficiently rigorous. It is recommended to supplement with a blocking experiment, i.e., pre-blocking by adding DNA sequences complementary to the target sequence before performing dfPP.

3.The phrase "data not shown" appears multiple times in the text (e.g., line 179, line 184, line 227). It is suggested to include these data in the supporting information.

4.In Figure 2F, the signal dots are difficult to discern. Please provide a clear, enlarged image with the locations of the signal dots circled.

5.In Table 1, the detection efficiency for the minimum target length is reported as 13%-65%. How is the detection efficiency defined? Please add the calculation method to the experimental procedures section. Could the large variation in detection efficiency (13%–65%) be due to the small sample size (only 4 males and 1 female)? Since the detection efficiency for DYZ1 and Pal3 can exceed 95%, does this indicate that dfPP is only suitable for detecting repetitive regions and not applicable for specific short target sequences?

**Do you want your identity to be public for this peer review?** For information about this choice, including consent withdrawal, please see our Privacy Policy

Reviewer #1: No

Reviewer #2: No

Reviewer #3: No

---

## [Author Response · Author response to Decision Letter 1]

29 Sep 2025

PLOS One

Dear Reviewers:

We are grateful for the insightful and constructive feedback for our manuscript. We have addressed all the points raised and revised the manuscript as detailed below.

We look forward to hearing from you at your earliest convenience.

Sincerely,

Ryoyo Ikebuchi and Ulf Landegren

Molecular Tools and Functional Genomics, Department of Immunology, Genetics and Pathology, Uppsala University

ryoyo.ikebuchi@igp.uu.se ulf.landegren@igp.uu.se

Uppsala Biomedical Center C10:2 Husargatan 3, Uppsala, Sweden, Box 815, SE-751 08

In this manuscript, the authors describe a denaturation-free method for in situ genomic DNA detection based on padlock probes and rolling-circle amplification (RCA). Short-lived openings in the DNA duplex are captured by circularizing oligonucleotides—padlock probes—that are enzymatically sealed by a DNA ligase in a sequence-specific manner. Once circularized and coiled around their target strands, the probes are amplified via RCA, yielding localized fluorescent signals for high-resolution visualization. Using this approach, genomic regions with thousands of repeats were detected in human leukocytes within a few hours, achieving over 99% efficiency and below 0.15% false positives. I think this work would meet the criteria of PLOS ONE. if the following issues are properly addressed.

1.Regarding the organization of the manuscript, Figure 1 presents actual sample detection results, while Figure 2 shows methodological validation. It is recommended to place the content of Figure 1 after Figure 4.

We have revised the manuscript in line with the reviewer’s comment. Figures 2, 3 and 4 in the original draft have been renumbered as Figure1, 2 and 3 in the revised version, while the original Figure 1 is now Figure 4. Additionally the data regarding dfPP specificity (originally presented as Figure 1H) has been relocated to Figure 3E to improve readability.

2.In Figure 2, the validation of DYZ1 specificity is not sufficiently rigorous. It is recommended to supplement with a blocking experiment, i.e., pre-blocking by adding DNA sequences complementary to the target sequence before performing dfPP.

Unfortunately, blocking experiments are not applicable to our assay, unlike RNA FISH, because the blocking oligonucleotide cannot access the double stranded target region (as shown in experiments where padlock probes were added in the absence of a ligase (new Fig. 1F left). Therefore, we have revised the sentence from “were indeed DYZ1-specific signals” to “were indeed chrY-specific signals” (Lines 128-129).

During this project, we considered performing FISH assay alongside dfPP to demonstrate padlock probe specificity by showing co-localization of FISH and dfPP signals. But the two assays cannot be combined because dfPP cannot be performed on cells that have undergone the denaturation steps required for FISH assays, as shown in Figure 1B, C and supporting figure 1A. This limitation motivated us to measure the spatial distances between signals from p22.31 and DXZ1 (Fig. 3E) to support the dfPP specificity.

3.The phrase "data not shown" appears multiple times in the text (e.g., line 179, line 184, line 227). It is suggested to include these data in the supporting information.

As suggested, we have included the imaging data related to denaturation conditions and ligation temperatures as supporting figure 1. Regarding FISH data, instead of presenting our own results, we have cited Reference 25 (Wilson et al., 2025), which provides high-quality karyotype data demonstrating within-culture mosaicism, including variations in chrY.

4.In Figure 2F, the signal dots are difficult to discern. Please provide a clear, enlarged image with the locations of the signal dots circled.

Following the reviewer’s suggestion, we have inserted white circles in enlarged images in Figure1G (Figure 2F in the original version) to improve clarity.

5.In Table 1, the detection efficiency for the minimum target length is reported as 13%-65%. How is the detection efficiency defined? Please add the calculation method to the experimental procedures section.

We have added an explanation for how detection efficiency is defined in lines 485-487, “The numbers of RCA products was then divided by the number of nuclei to calculate rate of false and correct positive signals in negative and positive samples, respectively”.

Additionally, we have included the average detection efficiency in line 230 and in Table 1, instead of presenting only the range of detection efficiencies.

Could the large variation in detection efficiency (13%–65%) be due to the small sample size (only 4 males and 1 female)?

We have added one more data point from a male sample, labeled as M5, in the revised Fig.3. The total sample size for TMSB4Y detection is now 5 males and 4 females (line 230-231). The M5 sample showed a detection efficiency of 32.4%, but a considerable variation remains with this larger number of investigated subjects. Additionally, we observed variability in the intensity of individual RCA signals from single copy targets in any samples. These findings suggest that unknown factors may still be influencing RCA kinetics in cell samples, as discussed in the third paragraph in Discussion.

Since the detection efficiency for DYZ1 and Pal3 can exceed 95%, does this indicate that dfPP is only suitable for detecting repetitive regions and not applicable for specific short target sequences?

As suggested, there are still limitations in detecting single copy targets using dfPP, likely due to lower detection efficiency against dsDNA-targets compared to ssDNA (Fig1G). Additionally, RCA may be compromised in tightly condensed chromatin (Fig. 3 and other publications [1,2]). However, these limitations can be overcome in certain contexts. As the reviewers suggests, and illustrated in the paper, detection of repetitive regions provides more opportunities for detection. As described in the Discussion (from line 378), application of dfPP for tracing clonal evolution in somatic tissues will involve monitoring multiple mutant cells compensating for the limited efficiency, and the technology also allows simultaneous targeting of several mutations to enhance detection efficiency for malignant cells. In Table. 1 we compare detection efficiency of dfPP to that of some of other DNA detection technologies. We note that dfPP has the advantage over other technologies that any sequence may be targeted as there are no requirement for specific sequence features and that mild dfPP procedure presented herein may prove advantages for simultaneous analyses of other, sensitive molecules in the samples (Line 328).

1. Yaroslavsky AI, Smolina I V. Fluorescence Imaging of Single-Copy DNA Sequences within the Human Genome Using PNA-Directed Padlock Probe Assembly. Chem Biol. 2013;20: 445–453. doi:10.1016/j.chembiol.2013.02.012

2. Zhang K, Deng R, Teng X, Li Y, Sun Y, Ren X, et al. Direct Visualization of Single-Nucleotide Variation in mtDNA Using a CRISPR/Cas9-Mediated Proximity Ligation Assay. J Am Chem Soc. 2018;140: 11293–11301. doi:10.1021/jacs.8b05309

---

## [Decision Letter · Decision Letter 1]

10 Oct 2025

Dear Dr. Ikebuchi,

Thank you for submitting your manuscript to PLOS ONE. After careful consideration, we feel that it has merit but does not fully meet PLOS ONE’s publication criteria as it currently stands. Therefore, we invite you to submit a revised version of the manuscript that addresses the points raised during the review process.

We look forward to receiving your revised manuscript.

Kind regards,

Ruijie Deng

Academic Editor

PLOS ONE

Journal Requirements:

Reviewers' comments:

Reviewer's Responses to Questions

**Comments to the Author**

Reviewer #1: All comments have been addressed

Reviewer #2: (No Response)

Reviewer #3: All comments have been addressed

2. Is the manuscript technically sound, and do the data support the conclusions?

Reviewer #1: Yes

Reviewer #2: Partly

Reviewer #3: Yes

3. Has the statistical analysis been performed appropriately and rigorously?

Reviewer #1: Yes

Reviewer #2: Yes

Reviewer #3: Yes

4. Have the authors made all data underlying the findings in their manuscript fully available?

Reviewer #1: Yes

Reviewer #2: No

Reviewer #3: Yes

5. Is the manuscript presented in an intelligible fashion and written in standard English?

Reviewer #1: Yes

Reviewer #2: No

Reviewer #3: Yes

Reviewer #1: There are some formatting issues in the current version of this manuscript. I suggest the author double-check carefully. On the other hand, the authors have addressed all my concerns. I recommend the publication of this paper.

Reviewer #2: The authors have addressed some previous concerns. However, some concerns still remain. This work is expected to meet the criteria for PLOS ONE, provided the following issues are adequately addressed.

1. The main text figures do not include bright-field (BF) images to confirm the morphology of the cells during imaging. It would strengthen the presentation to incorporate BF images and their corresponding fluorescence/BF merged images directly into the main figures to clearly demonstrate that the method is tested and functional in cells. This addition would enhance the readers' confidence to ensure that the cells are not getting damaged during the labeling.

2. The authors can use a simpler way of communicating their work here and without coming up with random words/terminologies. For example, “sequence resolution” in line 32.

3. Some fluorescence images lack clear scale bars.

Reviewer #3: The authors have adequately addressed my questions raised in the previous manuscript, and I think this manuscript is now acceptable. But, for publication, the author needs to add scale bar to each picture and provide the information in the captions.

**Do you want your identity to be public for this peer review?** For information about this choice, including consent withdrawal, please see our Privacy Policy

Reviewer #1: No

Reviewer #2: No

Reviewer #3: No

---

## [Author Response · Author response to Decision Letter 2]

13 Oct 2025

Journal Requirements:

We have carefully reviewed and corrected the reference list to ensure completeness and accuracy. Minor formatting issues (e.g., journal name and chapter citation style) have been addressed. No retracted articles are cited. All changes are reflected in the revised manuscript.

6. Review Comments to the Author

Reviewer #1: There are some formatting issues in the current version of this manuscript. I suggest the author double-check carefully. On the other hand, the authors have addressed all my concerns. I recommend the publication of this paper.

We sincerely thank Reviewer #1 for the positive evaluation and recommendation for publication. We appreciate your careful reading and helpful feedback. As suggested, we have thoroughly reviewed the manuscript for formatting issues and made the necessary corrections to ensure consistency and clarity. We have also double-checked the reference list and updated it where appropriate.

Reviewer #2: The authors have addressed some previous concerns. However, some concerns still remain. This work is expected to meet the criteria for PLOS ONE, provided the following issues are adequately addressed.

1. The main text figures do not include bright-field (BF) images to confirm the morphology of the cells during imaging. It would strengthen the presentation to incorporate BF images and their corresponding fluorescence/BF merged images directly into the main figures to clearly demonstrate that the method is tested and functional in cells. This addition would enhance the readers' confidence to ensure that the cells are not getting damaged during the labeling.

We thank Reviewer #2 for the valuable suggestion regarding BF images to confirm cell morphology. In our study, most fluorescence imaging experiments were performed without simultaneous acquisition of BF images, as our primary focus was on signal specificity and localization. However, to address your concern, we have included BF images corresponding to Figures 1B and 3D in Supporting Figures 1 and 4. These examples demonstrate that the method is functional in cells and that the observed signals are localized appropriately. We have updated the figure legends and referred to the Supporting Figures in the main text to guide the reader. To reflect this addition, we have also updated the final sentence of the first paragraph in the Results section to refer to the Supporting Figures and clarify the cellular context of the imaging.

2. The authors can use a simpler way of communicating their work here and without coming up with random words/terminologies. For example, “sequence resolution” in line 32.

As suggested, we have revised the wording in line 32 and replaced “sequence resolution” with a more precise and conventional expression. We have also reviewed the manuscript carefully to ensure that other terms are clearly defined and appropriately used (for example, line 283-284). If any remaining expressions appear unclear, we would be happy to revise them further.

3. Some fluorescence images lack clear scale bars.

We have reviewed all figures and added scale bars where they were missing. We have also updated the corresponding figure legends to include scale information.

Reviewer #3: The authors have adequately addressed my questions raised in the previous manuscript, and I think this manuscript is now acceptable. But, for publication, the author needs to add scale bar to each picture and provide the information in the captions.

We sincerely thank Reviewer #3 for the constructive feedback and positive assessment of our revised manuscript. In response to your suggestion, we have ensured that all relevant images now include clear scale bars, and the figure captions have been updated to reflect this information.

We hope that the revisions and clarifications provided here adequately address all remaining concerns. We thank the reviewers and editors again for their time and thoughtful feedback.

---

## [Editor Report · Decision Letter 2]

14 Oct 2025

A denaturation-free protocol for in situ visualization of short nuclear DNA sequences using padlock probes with rolling-circle amplification

PONE-D-25-37259R2

Dear Dr. Ikebuchi,

We’re pleased to inform you that your manuscript has been judged scientifically suitable for publication and will be formally accepted for publication once it meets all outstanding technical requirements.

Kind regards,

Ruijie Deng

Academic Editor

PLOS ONE
---

## [Editor Report · Acceptance letter]

PONE-D-25-37259R2

PLOS ONE

Dear Dr. Ikebuchi,

I'm pleased to inform you that your manuscript has been deemed suitable for publication in PLOS ONE. Congratulations! Your manuscript is now being handed over to our production team.

Kind regards,

on behalf of

Dr. Ruijie Deng

Academic Editor

PLOS ONE